# Color Night Light Remote Sensing Images Generation Using Dual-Transformation

**DOI:** 10.3390/s24010294

**Published:** 2024-01-03

**Authors:** Yanling Lu, Guoqing Zhou, Meiqi Huang, Yaqi Huang

**Affiliations:** 1Guangxi Key Laboratory of Spatial Information and Geomatics, Guilin University of Technology, Guilin 541004, China; luyl2014@glut.edu.cn; 2College of Geomatics and Geoinformation, Guilin University of Technology, Guilin 541004, China; 6614031@glut.edu.cn (M.H.); 2120221988@glut.edu.cn (Y.H.); 3College of Earth Sciences, Guilin University of Technology, Guilin 541004, China

**Keywords:** multi-source remote sensing, IHS color space transform, wavelet transform, image fusion, color night light remote sensing images

## Abstract

Traditional night light images are black and white with a low resolution, which has largely limited their applications in areas such as high-accuracy urban electricity consumption estimation. For this reason, this study proposes a fusion algorithm based on a dual-transformation (wavelet transform and IHS (Intensity Hue Saturation) color space transform), is proposed to generate color night light remote sensing images (color-NLRSIs). In the dual-transformation, the red and green bands of Landsat multi-spectral images and “NPP-VIIRS-like” night light remote sensing images are merged. The three bands of the multi-band image are converted into independent components by the IHS modulated wavelet transformed algorithm, which represents the main effective information of the original image. With the color space transformation of the original image to the IHS color space, the components I, H, and S of Landsat multi-spectral images are obtained, and the histogram is optimally matched, and then it is combined with a two-dimensional discrete wavelet transform. Finally, it is inverted into RGB (red, green, and blue) color images. The experimental results demonstrate the following: (1) Compared with the traditional single-fusion algorithm, the dual-transformation has the best comprehensive performance effect on the spatial resolution, detail contrast, and color information before and after fusion, so the fusion image quality is the best; (2) The fused color-NLRSIs can visualize the information of the features covered by lights at night, and the resolution of the image has been improved from 500 m to 40 m, which can more accurately analyze the light of small-scale area and the ground features covered; (3) The fused color-NLRSIs are improved in terms of their MEAN (mean value), STD (standard deviation), EN (entropy), and AG (average gradient) so that the images have better advantages in terms of detail texture, spectral characteristics, and clarity of the images. In summary, the dual-transformation algorithm has the best overall performance and the highest quality of fused color-NLRSIs.

## 1. Introduction

Compared with traditional remote sensing images acquired during the daytime, night light remote sensing images (NLRSIs) record the surface light intensity at night, which can reveal the potential pattern of human activities that cannot be observed in daytime images and has thus been widely applied in the estimation of socio-economic parameters such as GDP, electricity consumption, light pollution, poverty index, and Gini coefficient, etc. [1,2,3,4,5]. Current research on night light image processing focuses on the integration of Defense Meteorological Satellite Program/Operational Linescan System (DMSP/OLS) and National Polar-orbiting Partnership/Visible Infrared Imaging Radiometer (NPP/VIIRS) night light images to obtain high temporal resolution night images [6]. LuoJia1-01, the first visible hyper-spectral and nocturnal multi-spectral multi-mode programmable micro-nanosatellite in orbit in China [7], has overcome the limitations of traditional nocturnal remote sensing images (i.e., black and white images with low resolution) and provides a new perspective and data source for small-scale urban development research [8]. The NPP-VIIRS-like night light remote sensing images datasets are a cross-sensor correction scheme based on the auto-encoder (AE) model proposed by Yu’s team and can be directly used simultaneously with NPP-VIIRS night light images [9]. Most of the existing studies focus on using single NLRSIs, and the current NLRSIs have a long time span; problems such as relatively low resolution and wide research scope hinder the accurate representation of the distribution characteristics of human activities in small-scale areas. Thus, it is necessary to fuse multi-source images to generate color-NLRSIs with more accuracy and detail.

In the processing of NLRSI, researchers have proposed various cross-sensor calibration models to resolve the inconsistency between DMSP/OLS and NPP/VIIRS data. Li (2017) and Wu et al. (2019) fit DMSP/OLS and NPP/VIIRS data with a power function and Gaussian low-pass filtering to DMSP/OLS data [10,11]. Tu et al. (2020) proposed a novel cross-sensor calibration framework for DMSP/OLS and NPP/VIIRS data to effectively overcome the shortcomings of DMSP/OLS data and generate night light images with a long time series [12]. Chen and Yu et al. (2020) proposed a cross-sensor correction scheme based on the AE model to develop a new long time-series (2000–2020) night light image with high accuracy [9]. Notably, multi-band fusion for night light remote sensing images is still experimental, whereas multi-source remote sensing image fusion involves many fields. Multi-source remote sensing image fusion is an algorithm designed based on practical problem requirements and applications. With improvements in computer performance, hardware technology, and wavelet transforms in recent years, image fusion by wavelet transformation has become a popular research topic [13]. Ranchin et al. (1993) were the first to apply wavelet transform to the field of remote sensing image fusion [14]. Do et al. (2003) proposed image fusion of the wavelet transform of multi-resolution analysis (MRA) and the contourlet transform of time-frequency localization (TFL) based on the wavelet transform to overcome the shortcomings of the wavelet transform [15]. Maria et al. (2004) improved the fast intensity hue saturation (IHS) fusion algorithm to reduce the spectral distortion produced during fusion [16]. Zhou et al. (2014) introduced a generalized IHS algorithm for fusing more than two sensor images [17]. Liu et al. (2018) combined multi-scale analysis to optimize the processing of different wavebands and proposed an improved Gram-Schmidt (GS) fusion algorithm based on the IHS transform [18]. Wang et al. (2021) combined the advantages of three algorithms; namely, IHS, Principal Component Analysis (PCA), and wavelet transform, to minimize the spectral distortion caused by uncorrelated replacement components and resolve the detailed distortion of wavelet transform [19].

Notably, the fused NLRSIs are generally a fusion of optical images and night light images, and the intensity and morphological information of night light images are fused by comprehensively using the texture and spectrum characteristics of optical images to extract the potential information of human activities. However, owing to numerous complicated factors in the process of image fusion, fused images generally have many problems, such as spectral distortion and spatial detail distortion [20,21]. Despite the rapid development of remote sensing technology, existing research results cannot meet practical requirements, especially when different sensors are fused [22,23].

Although NLRSIs are receiving increasing attention, the application of research still has certain limitations. Integrating multi-sensor, multi-scale time-phase remote sensing images to obtain high-resolution color-NLRSIs to alleviate “spatial-temporal conflict” is imminent. To address the above problems, in this study, we expand the research space of the multi-source remote sensing image fusion, completely use the complementary information between night light remote sensing and other remote sensing images, and further improve the spatial-temporal and spectral resolutions of images through fusion algorithms. The image information of the Terra, OLS, and VIIRS sensors is fully fused, and the dual-transformation algorithm is used to conduct multi-band synthesis of night light remote sensing images to generate color-NLRSIs, aimed at improving the resolution of night light remote sensing images to widen the application potential.

## 2. Methods

### 2.1. Dual-Transformation for Fused Images

The IHS transform fusion algorithm is a transform based on the IHS color space. When processing an image, the RGB channel of the source image is separated into IHS channels (three elements of intensity I, hue H, and saturation S). This integrates the RGB image information by transforming components I, H, and S to fuse images with different spectral ranges. In the IHS transform process, we retain the detail of the original image, H and S represent the color information of the original image. The multi-spectral image is IHS transform to separate the RGB channel from the IHS channel, as Equations (1)–(3) show [24]:(1)$$H=\arctan\left(\frac{S_{1}}{S_{2}}\right)$$
where H denotes the hue feature components in IHS transformation, $S_{1}$ and $S_{2}$ denote the intermediate variables of RGB conversion to the IHS color space.
(2)$$S=\sqrt{\left(S_{1}^{2}+S_{2}^{2}\right)}$$
where S denotes the saturation feature components in IHS transformation, $S_{1}$ and $S_{2}$ denote the intermediate variables of RGB conversion to the IHS color space.
(3)$$\left[\begin{array}{c} I\\ S_{1}\\ S_{2} \end{array}\right]=\left[\begin{array}{ccc} \frac{1}{\sqrt{3}} & \frac{1}{\sqrt{3}} & \frac{1}{\sqrt{3}}\\ \frac{1}{\sqrt{6}} & \frac{1}{\sqrt{6}} & \frac{-2}{\sqrt{6}}\\ \frac{1}{\sqrt{2}} & \frac{-1}{\sqrt{2}} & 0 \end{array}\right]\left[\begin{array}{c} R\\ G\\ B \end{array}\right]$$
where I denotes the intensity feature components in IHS transformation; $S_{1}$ and $S_{2}$ denote the intermediate variables of RGB conversion to the IHS color space; R, G, and B denote the red, green, and blue bands of the multi-spectral image, respectively.

As both the component  I and the panchromatic image can reflect the gray change in ground objects, the component  I is expressed in the panchromatic image to obtain the component $I_{new}$ with more detailed features. Finally, the new RGB image is obtained using the IHS inverse transformation, as shown in Equation (4) follows [25]:(4)$$\left[\begin{array}{c} R_{\mathrm{new}}\\ G_{new}\\ B_{new} \end{array}\right]=\left[\begin{array}{ccc} \frac{1}{\sqrt{3}} & \frac{1}{\sqrt{6}} & \frac{1}{\sqrt{2}}\\ \frac{1}{\sqrt{3}} & \frac{1}{\sqrt{6}} & \frac{-1}{\sqrt{2}}\\ \frac{1}{\sqrt{3}} & \frac{-2}{\sqrt{6}} & 0 \end{array}\right]\left[\begin{array}{c} I_{new}\\ S_{1}\\ S_{2} \end{array}\right]$$
where $R_{new}$, $G_{new}$ and $B_{new}$ denote the red, green, and blue new bands of the multi-spectral image; $S_{1}$ and $S_{2}$ denote the intermediate variables of RGB conversion to the IHS color space; $I_{new}$ denotes the new value of I after the IHS transformation.

The process of image fusion based on IHS transform involves the component replacement in the IHS color space, decomposition of the multi-spectral image, and obtaining the component I of NLRSI. Both component I  and panchromatic images can reflect the grayscale variation of features and have similar spatial texture features. Therefore, the panchromatic image is used to substitute the component I  to obtain the component $I_{new}$ with more geometric structures. Finally, the IHS inverse transform is performed to obtain a new RGB fused image, as shown in Figure 1.

An MS image contains multi-spectral bands, and detailed information on surface features can be obtained from an MS image by analyzing the spectral information of the different bands. IHS transform can separate the brightness, hue, and saturation of the image, thus effectively retaining the color information and color distribution without causing color distortion and retaining the spatial structure information of the image to a greater extent. IHS transform has the advantages of easy operation and fast performance compared with other fusion methods. Meanwhile, it can effectively inject the detailed information of the PAN image into the MS image so that the spatial structure information can also be retained to a larger extent.

However, IHS transform only processes multi-spectral images within three bands. If the IHS transform is still used in multi-band images, spectral degradation occurs. This is mainly due to the loss and fuzzy spectral information in the process of IHS transformation, which affects the accuracy of analysis of remote sensing data and cannot meet the processing of variable remote sensing data. In concrete terms, the spectral degradation is mainly due to the linear transformation of the light components during the IHS transform. In IHS transform, the light information of the MS image is acquired by linear transformation fused with the light information of PAN images. This linear transformation leads to compression or loss of spectra in the original MS image, thus causing spectral degradation. Therefore, the IHS transform must integrate other algorithms to conduct multi-band image fusion better.

Wavelet transform is a multi-scale signal analysis and reconstruction method that can decompose the image into sub-band channels of different frequencies [26,27]. It can also extract detailed information about the image in the sub-bands of different frequencies, especially the high-frequency sub-bands, to make up for the detailed information that may be lost during the IHS transform. It can be seen that the dual-transformation algorithm, which integrates the wavelet and IHS transform, can better solve the problem of spectral degradation of the IHS transform. In addition, the spectral information of the original image can also be better preserved in the fusion process, making the fused image closer to the spectral characteristics of the original image.

The specific process is described as follows:

(1) The wavelet coefficient matrices in different directions can be obtained by separately performing wavelet decomposition on the low-resolution MS images and high-resolution PAN images to be merged. They contain the feature information of the images at different scales and frequencies. One-dimensional wavelet decomposition is obtained for high-frequency component *W* and low-frequency component *L* in the horizontal direction. Two-dimensional wavelet decomposition is obtained for low-frequency and high-frequency components in the horizontal and vertical directions [28,29], as expressed in Equations (5) and (6). Equation (5) denotes the coefficient matrix of wavelet decomposition, and the values reflect the weight distribution of the different frequency components. The center of value is the highest and decreases in all directions, which is used to extract information at different scales from the original image, as follows:(5)$$\frac{1}{256}\left[\begin{array}{ccccc} 1 & 4 & 6 & 4 & 1\\ 4 & 16 & 24 & 16 & 4\\ 6 & 24 & 36 & 24 & 6\\ 4 & 16 & 24 & 16 & 4\\ 1 & 4 & 6 & 4 & 1 \end{array}\right]$$

Equation (6) denotes the low-pass decomposition operator as follows:(6)$$W_{P}=P_{L}-P_{LL},W_{T}=T_{L}-T_{LL}$$
where $P_{L}$ and $T_{L}$ denote the low-frequency components of PAN images $\left(P\right)$ and MS images $\left(L\right)$; $P_{LL}$ denotes the low-frequency components of the $P_{L}$; $W_{P}$ denotes the high-frequency components of the $P_{L}$; $T_{LL}$ denotes the low-frequency components of the ${\;T}_{L}$; $W_{T}$ denotes the high-frequency components of the ${\;T}_{L}$.

(2) The wavelet coefficients from each layer decomposition are fused using different fusion rules according to the characteristics of different frequency components. For high-frequency wavelet coefficients, fusion is achieved by fusing with the absolute maximum; for low-frequency wavelet coefficients, fusion is achieved by fusing with the average value or variance. Using this method, the feature information of the MS image and PAN image can be fully utilized so that the fused image has not only the rich information and spatial distribution characteristics of the MS image but also the high spatial resolution and detailed information of the PAN image. Wavelet transform has an excellent ability to reconstruct images by decomposing them into detailed and averaged components of the source image. Methods such as the absolute maximum of coefficients, weighted average, and local variance criterion are commonly used to determine the fusion coefficients. The method of maximum absolute coefficient is used to determine the fusion coefficient by finding the pixel value that has the maximum influence which can achieve better fusion; the method of weighted average is used to calculate the fusion coefficients by providing weights for different images; the weights can be adjusted according to the characteristics and application requirements of images; the method of local variance criterion determines the importance of image details in the fusion process by calculating the friction prevention within a local window. Thus, the methods of fusion coefficient have their own advantages, and the best method should be selected according to the actual situation.

In this study, the method of the maximum absolute coefficient and weighted average was used to generate the fusion coefficients. Weighted analysis of low-frequency components was based on different fusion coefficients so that low-frequency components of MS images could be reconstructed. This is expressed in Equation (7):(7)$${T^{\prime }}_{LL}=K_{T}\times T_{LL}+K_{P}\times P_{LL}$$
where $K_{T}$ and $K_{P}$ denote the weighting coefficients and ${T^{\prime }}_{LL}$ denotes the low-frequency component of the reconstruction when $K_{T}{>K}_{P}$; $T_{LL}$ denotes the low-frequency components of the ${\;T}_{L}$; and $P_{LL}$ denotes the low-frequency components of the $P_{L}$. The variation in weighting coefficients affects the contribution of the images in the fusion result; therefore, the fusion result can be adjusted by revising the weighting relationship between different images; the weights of different images at each pixel can be determined by transforming the fusion coefficients, thus affecting the changes in the spectral or spatial features of the fused image.

(3) The high-frequency detail components of the MS and PAN images were replaced.

(4) All fused wavelet coefficient matrices are inverted using wavelet inversion to obtain the fused image, and the results of steps (2) and (3) were used for inverse wavelet transform, as shown in Figure 2.

### 2.2. Color-NLRSIs Generation

Although the conventional IHS color transform preserves as much information as possible, such as the hue and saturation of multi-spectral images and spatial detail features of panchromatic images, the fusion process generally suffers from spectral and spatial information distortions [30,31]. Traditional wavelet transform decomposes the image into different frequencies for selective fusion, which greatly maintains the information of spectral and spatial structure features of the image [32,33]; however, certain problems inherent in the high complexity of algorithms still exist, such as the number of wavelet decomposition layers and wavelet basis selection. Therefore, the use of the IHS color space modulation wavelet transform for the multi-band fusion of night light remote sensing images can overcome the problems of single band and low resolution of traditional night light remote sensing images. It can also enhance the advantages of both algorithms while improving the spatial resolution of the fused images. Multi-spectral (MS) images with different spatial resolutions are fused with PAN images, and detailed information on basis functions is selected from the simplest Haar wavelet function in the two-dimensional discrete wavelet transform. The specific steps are as follows.

(1) Image pro-processing. Landsat MS images (30 m) and PAN images (15 m) are aligned and resampled. Alignment is the process of selecting control points on the MS image and aligning them to the PAN image for reference; thus, they correspond to the same spatial position in the harmonized coordinate reference and ensure that they match each other in size, orientation, and position. Compared with NLRSI, the MS image (30 m) and the PAN image (15 m) in Landsat have a better spatial resolution. As the difference between Landsat and NLRSI resolutions is large, it is necessary to up-sample the Landsat and down-sample the NLRSI. It is calculated that a 40 m resolution is the smallest integer multiple of a 500 m resolution (NLRSI), so the experiment used 40 m as the standard for image resampling. This enables simultaneous up-sampling of Landsat and down-sampling of NLRSI. Therefore, the MS and PAN images were resampled to 40 m × 40 m to harmonize their size and dimensions.

(2) IHS positive transformation. Image processing is usually displayed in RGB, and the R, G, and B channels of MS images are positively transformed in the IHS color space to extract the components I, H, and S of MS images. According to the conversion relationship between the IHS and RGB models, the MS image is IHS transformed; the image is transformed to the IHS color space; the original image is decomposed into three channels, namely, R, G, and B; and the component I containing the detail information and components H and S containing the spectral information are extracted in the IHS color space. Components H and S of the MS image contain spectral information, such as color and saturation, and component I contains detailed information, such as the spatial structure and features [34], as shown in Figure 3.

(3) Image histogram matching. After IHS color space transformation, components H and S of the MS image remain unchanged, and component I of the MS image with spatial details is used for histogram matching with the PAN image containing richer spatial details; the grayscale value of the PAN image is then calculated, and the histogram is equalized with the component I of the MS image, as expressed in Equation (8). According to the correspondence between $Z_{K}$ and $P_{Z}$, the gray level of the PAN image is adjusted to obtain $I_{new}$ with a higher degree of matching with the original image and $I_{new}$ is matched with the PAN image histogram to obtain ${PAN}_{new}$ [35,36].
(8)$$S_{r}\left(Z_{k}\right)=\sum_{j=0}^{k}P_{z}\left(Z_{j}\right)\;(k=0,1,\ldots ,L-1)$$
where $Z_{K}$ denotes the PAN image grayscale value, $P(Z_{K})$ denotes the probability estimate, and $S_{r}(Z_{K})$ denotes the sum of $P(Z_{K})$.

(4) Two-dimensional wavelet decomposition. The component I of the MS image and PAN image are wavelet transforms. The wavelet transform (row and column) of the two images is a one-dimensional wavelet decomposition, and the low-frequency component L and the high-frequency (detail) component H of the image are obtained by row decomposition. The low-frequency coefficient and detail coefficient of the image are obtained by column decomposition based on row decomposition. In particular, the image is decomposed into four sub-components containing the effective information regarding the I and P of the original MS image, namely, the low-frequency sub-component LL of the original image, the horizontal detail feature LH, the vertical detail feature HL, and the diagonal feature HH [37]. The low-frequency sub-component LL is decomposed by the two-dimensional wavelet; namely, the image information is analyzed by multi-resolution transformation, and the high- and low-frequency coefficients of the image are separated, which is convenient for extracting image features, as shown in Figure 4.

(5) Selection of fusion coefficients and wavelet reconstruction. Components $I_{new}$ and ${PAN}_{new}$ are decomposed into four low-frequency and detail components in different directions (horizontal and vertical) by a two-dimensional discrete wavelet transform, which contains the effective information of the image [38]. We adopted the coefficient absolute value larger method and selected the appropriate fusion coefficient weights according to different fusion rules. We used the absolute value strategy for the detail component and the mean strategy for the low-frequency component, as expressed in Equations (9) and (10).
(9)$$C\left(F,p\right)=\left\{\begin{array}{l} W_{\max}C\left(A,p\right)+W_{\min}C\left(B,p\right),G\left(A,p\right)\geq G\left(B,p\right)\\ W_{\min}C\left(A,p\right)+W_{\max}C\left(B,p\right),G\left(A,p\right)<G\left(B,p\right), \end{array}\right.$$
(10)$$C\left(F,p\right)=\left\{\begin{array}{c} C\left(A,p\right),\left|C\left(A,p\right)\right|\geq \left|C\left(B,p\right)\right|\\ C\left(B,p\right),\left|C\left(A,p\right)\right|<\left|C\left(B,p\right)\right| \end{array}\right.$$
where Equations (9) and (10) denote the low- and high-frequency fusion rules, respectively; C(F,p) denotes the spatial coefficient matrix of image F in the p domain; $W_{max}$ and $W_{min}$ denote the maximum and minimum weights, respectively; and p=(m,n) denotes the spatial location of the coefficient matrix. Therefore, different fusion coefficients are assigned according to the high- and low-frequency components of wavelet decomposition, and the high- and low-frequency coefficients obtained in step (4) are wavelet-inverted to complete the reconstruction of component I of the MS and PAN images to obtain $I_{new}$.

(6) IHS inverter. The extracted high- and low-frequency detail components were fused by wavelet reconstruction and the inverse wavelet transform to obtain the wavelet transformed $I_{new}$. The obtained $I_{new}$ is inverted with H and S obtained in step (2) via IHS inverse transform to obtain the fused image, as shown in Figure 5.

From this, the MS (R, G, B) images were obtained by multi-band fusion of Landsat_MS and NLRSI red-green bands. Then, they were converted from MS (R, G, B) to MS (I, H, S) by IHS positive transform and component $I_{new}$ was obtained by image histogram matching, wavelet decomposition, and reconstruction between Landsat_PAN and component  I. The $I_{new}$ and MS (H, S) were fused, and then IHS inverse transformation was performed to fuse the image (R, G, B), that is, color-NLRSI (R, G, B). The entire process of the dual-transformation method for generating color-NLRSI is shown in Figure 6.

## 3. Experiments and Analysis

### 3.1. Study Area and Data Pro-Processing

#### 3.1.1. Study Area

The study area is located in the city of Shanghai, China. Shanghai is located in the southeastern coastal region of China, on the southern flank of the Yangtze River, adjacent to the provinces of Jiangsu and Zhejiang, at 120°52′ E–122°12′ E, 30°40′ N–31°53′ N, with a land area of about 6300 km^2^. Shanghai is a socio-economically well-developed central city in the country. In recent years, the rapid urban development in Shanghai and the fast expansion of the city center have been conducive to studying night light remote sensing.

#### 3.1.2. Datasets and Pro-Processing

This study mainly used NPP-VIIRS-like night light remote sensing images, Landsat MS images, and PAN images. These images were fused to generate high-quality color night light remote sensing images by complementing each other’s advantages. Traditional night light remote sensing image is a black and white image with low resolution and a single band, which means that the change in night light remote sensing data between months is not obvious enough. Therefore, to make the changes in night light in Shanghai over 9 years more visible, this study selected study data spanning across years. The specific datasets used are as follows:

(1) The NPP-VIIRS-like night light remote sensing datasets spanning from 2000 to 2020, which were derived from the new long time-series (2000–2020) night light datasets proposed and established by Yu’s team with a spatial resolution of 500 m [9,19]; and the NPP-VIIRS-like data were extracted for the Shanghai study area.

(2) The Landsat MS and PAN datasets, which had spatial resolutions of 30 and 15 m for Shuttle Radar Topography Mission (SRTM) V3 data. Landsat-7 ETM+ and Landsat-8 OLI_TIRS were primarily used to obtain remote sensing images of Shanghai from 2000 to 2020, and these data were obtained from the Geospatial Data Cloud Platform (http://www.gscloud.cn. Accessed on 20 May 2021.). Considering the strip-missing failure and continuous cloud contamination of Landsat 7, 22 scenes of Landsat data with cloud volume below 10% were chosen as valid data. Thus, we selected Landsat-7ETM SLC-on data for 2000 and 2002 with strip numbers 118-038 and 118-039, respectively; Landsat-7 ETM SLC-off data for 2004, 2006, 2008, and 2010, Landsat-7ETM SLC-off data for 2012; and the Landsat 8 OLI_TIRS images for 2014, 2016, 2018, and 2020, respectively. Landsat-7 ETM+ and Landsat-8 OLI_TIRS were used to acquire remote sensing images of Shanghai from 2000 to 2020. The study area is shown in Figure 7.

Because of the difference between the datum, spatial resolution, atmospheric conditions, etc., the datasets were all resampled to the same spatial resolution using a triple convolutional interpolation method; the atmospheric correction method was used for the Landsat_MS datasets (see Figure 8), and the radiometric calibration method was used for the Landsat_PAN datasets (see Figure 9). The processing results are shown in Figure 8, Figure 9 and Figure 10.

### 3.2. Color-NLRSIs Generations in Shanghai from 2000 to 2020

Considering that multi-source image fusion is a pixel-wise fusion, the source images need to be aligned and sampled to maintain consistent pixel widths, including the alignment and resample of Landsat images of the Shanghai study area. We conducted an IHS color space transformation to obtain the component I containing the detail information and the components H and S containing spectral information in the IHS color space. Then, we matched $I_{new}$ with the PAN image histogram to obtain ${PAN}_{new}$. We conducted two-dimensional wavelet decomposition to separate the high- and low-frequency coefficients of the image to facilitate the extraction of image features; the fusion coefficients and wavelet reconstruction were selected to reconstruct the component I of MS and PAN images. The fusion of the multi-source images is shown in Figure 11.

Based on the spectral differences between daytime optical images and black-and-white night light images, the night light remote sensing images were single-band images, whereas the daytime optical images included multi-spectral bands. The red and green bands of Landsat multi-spectral images were fused with night light remote sensing images using dual-transformation to generate color night light remote sensing images. As the NLRSIs are single-band images, the night light images of Shanghai can be fused with Landsat multi-spectral data for band fusion to generate the color-NLRSIs. The results are shown in Figure 12.

### 3.3. Comparisons Analysis

To examine the effectiveness of the proposed method, the PCA, IHS, Wavelet, Brovey, and dual-transformation (IHS + WT) were used for comparison analysis. The results were analyzed by comparing their processing results in different images using the subjective evaluation indexes. Figure 13 shows the comparison of different fusion algorithms for Landsat images in the study area, and Figure 14 shows the comparison of different algorithms for color-NLRSIs. The dual-transformation algorithm in this study is defined as IHS + WT (see Figure 13 and Figure 14). The image fusion of Landsat and night light remote sensing using the IHS-modulated wavelet transform algorithm produces a color-NLRSI whose spatial resolution is improved to 40 m compared with that of the original night light remote sensing (500 m). This image has the best overall performance in terms of detail contrast and color information before and after fusion and the best color-NLRSI quality.

The comparison of spatial details using different algorithms for fusion in Landsat_MS, is shown in Figure 15, shows great improvement compared with the original multi-spectral image. Notably, the algorithm of IHS adjusted wavelet transform is clearer in the red rectangular frame area. In Figure 15f, the buildings on both sides of the road near the Bird’s Nest are slightly blurred visually, and mosaics are present near the Bird’s Nest. The visual clarity of Figure 15c–e are consistent, but the color information of Figure 15d is not consistent with that of the original multi-spectral image. The overall color tone shown in Figure 15e is dark. In summary, the dual-transformation algorithm used in Figure 15h has the best effect on the details.

The comparison of spatial details using different algorithms for fusion in Landsat_PAN is shown in Figure 16. The red rectangular box area shows that the fusion results of all five algorithms are visually clearer; however, the overall tone of Figure 16e is dark, and the color information in Figure 16d,f,g is not consistent with that of the original multi-spectral image. In summary, the dual-transformation algorithm used in Figure 16h has the best effect on detail, with clearer texture information and richer color information, which is more consistent with the visual system of the human eye.

The comparison of spatial details using different algorithms for fusion in color-NLRSIs is shown in Figure 17. Only Figure 17f is visually blurred, and the fusion results of the remaining five algorithms are visually clearer. However, the overall tone of Figure 17e is darker, the color information in Figure 17d,g is not consistent with that of the original multi-spectral image, and the intensity of Figure 17c is slightly greater. In summary, the dual-transformation algorithm used in Figure 17h had the best effect on detail, and the image was clearer visually.

### 3.4. Evaluation of Image Quality Metrics

The following indicators are used for objective evaluation of fusion image quality.

(1) Information entropy (*EN*) indicates the richness of the image information and indicates the average amount of information in the image. It can be expressed as follows [39]:(11)$$EN\left(F\right)=\sum_{i=1}^{m}Z_{i}lbZ_{i}$$
where $Z_{i}=\left\{Z_{1},Z_{2},\cdots ,Z_{m}\right\}$ denotes the gray distribution, $Z_{i}$ denotes the first-level gray probability density, i denotes the degree, and m denotes the gray level.

(2) The average gradient (*AG*). It can be used to evaluate the clarity of the image while reflecting the small detail contrast and texture transformation characteristics in the image, indicating the image’s sharpness [40]. It can be expressed as follows:(12)$$AG=\frac{1}{\left(M-1\right)\left(N-1\right)}\times \sum_{i=1}^{M-1}\sum_{j=1}^{N-1}\sqrt{\frac{{\left(F\left(i+1,j\right)-F\left(i,j\right)\right)}^{2}+{\left(F\left(i,j+1\right)-F\left(i,j\right)\right)}^{2}}{2}}$$
where M and N denote the width and height of image F, respectively, and F(i,j) denotes the grayscale value of image F at (i,j).

(3) The mean value (*MEAN*) indicates the average intensity information of the image [41] and reflects the average intensity of the image; the more moderate the value, the better the fusion effect. It can be expressed as follows:(13)$$\bar{F}=\frac{1}{MN}\sum_{i=1}^{M}\sum_{j=1}^{N}F\left(i,j\right)$$

(4) The standard deviation (*STD*) indicates the degree of image contrast [42] and reflects the dispersion of the image grayscale relative to the *MEAN*. It can be expressed as follows:(14)$$STD=\sqrt{\frac{1}{MN}\sum_{i=1}^{M}\sum_{j=1}^{N}{\left[F\left(i,j\right)-\bar{F}\right]}^{2}}$$
where $\bar{F}$ denotes the grayscale average of the image.

(5) The root-mean-square error *(RMSE)* indicates the amount of spatial detail feature information, and the smaller the value, the better the fusion effect [43]. It can be expressed as follows:(15)$$RMSE\left(I_{F},I_{W}\right)=\sqrt{\frac{1}{MN}\sum_{i=1}^{M}\sum_{j=1}^{N}{\left(I_{F}\left(i,j\right)-I_{W}\left(i,j\right)\right)}^{2}}$$
where $I_{F}$ denotes the fused image and $I_{W}$ denotes the source image.

(6) The correlation coefficient (*CC)* indicates the correlation between the fused and source images [43]; the closer the value of the correlation coefficient to 1, the greater the correlation. It can be expressed as follows:(16)$$CC\left(I_{F},I_{W}\right)=\frac{\sum_{i=1}^{M}\sum_{j=1}^{N}\left(I_{F}\left(i,j\right)-\bar{I_{F}}\right)\left(I_{W}\left(i,j\right)-\bar{I_{W}}\right)}{\sqrt{\sum_{i=1}^{M}\sum_{j=1}^{N}{\left(I_{F}\left(i,j\right)-\bar{I_{F}}\right)}^{2}\times \sum_{i=1}^{M}\sum_{j=1}^{N}{\left(I_{W}\left(i,j\right)-\bar{I_{W}}\right)}^{2}}}$$

Among them, the dual-transformation has the largest *EN* value, indicating that the color-NLRSIs contain richer information, with a *CC* value of 0.867, which is the closest to 1 among the above five algorithms.

Thus, the results of the dual-transformation algorithm are shown as the maximum *AG* value, indicating that it has high image clarity and facilitates the visual interpretation of the images. The *RMSE* value was relatively minimal, and the spatially detailed information was stable. The *STD* was the largest, and the contrast of the image results before and after fusion was high. The *MEAN* was moderate, and the mean value of the fused image was stable. The results are shown in Figure 18.

With regard to the three types of images (Landsat_MS/PAN and color-NLRSIs), Figure 19a (comparison of the results of different algorithms for Landsat_MS images) shows that the *MEAN*, representing the average intensity information of the image and its value is between 80 and 100; a more moderate value implies a better fusion effect. The *RMSE* in the IHS algorithm is slightly lower than that of the PCA algorithm, and the smaller the value of the *RMSE*, the better the fusion effect. The values of *MEAN*, *STD*, *CC*, and *EN* are better than those of the PCA algorithm, and the larger their values, the better the fusion effect. This indicates that the IHS algorithm images are of better quality regarding their clarity and information-carrying capacity. The *MEAN* of the Brovey algorithm is 28.483, which is too low, and the *AG* and *CC* values are smaller than those of other algorithms, indicating that the Brovey algorithm has the worst fusion effect. The *EN* of the wavelet algorithm is 0.025, which is a small decrease compared with the other algorithms, but the remaining descriptive index values were improved; thus, the wavelet algorithm is better than the above three algorithms. The *AG* value of the dual-transformation algorithm is 14.956, and the *EN* is 0.069; both are larger than those of the wavelet algorithm. Thus, the image detail contrast description and correlation degree are better than those of the wavelet algorithm, and the image fusion quality is the best.

Figure 19b (a comparison of the results of different algorithms for Landsat_PAN images) shows that the *MEAN* of all algorithms is moderate, between 110 and 130; the more moderate the mean value, the better the value of the fusion effect. The IHS algorithm is better than the PCA algorithm in terms of the *STD*, *CC*, and *EN*. The PCA algorithm is better than the IHS algorithm in terms of reflecting the image detail features and sharpness, but the *RMSE* is 24.093, which is greater than that of the IHS algorithm; the smaller the *RMSE* value, the better the fusion effect. The PCA algorithm shows severe spectral distortion, and the image color information changes before and after fusion; thus, the IHS algorithm is better than the PCA algorithm. The IHS algorithm is better than the Brovey algorithm in terms of the *MEAN*, *STD*, *AG*, and *CC*. The *STD* indicates the degree of image contrast, and the AG indicates the image sharpness; the larger their values are, the better the fusion effect. The *MEAN* value of the Brovey algorithm shows a serious decreasing tendency and is only 42.585; thus, the IHS algorithm is better than the Brovey algorithm. The wavelet algorithm outperforms the IHS algorithm regarding both objective quality-evaluation metrics. In the dual-transformation algorithm, the *MEAN* and *RMSE* are 116.945 and 19.111, respectively; both index values are slightly inferior to those of the wavelet algorithm, and the remaining indexes are better than those of the wavelet algorithm; thus, the dual-transformation algorithm has the best image fusion quality.

Figure 19c (a comparison of the analysis results of different algorithms for color-NLRSIs) shows that the *MEAN* values of the five algorithms are basically between 40 and 50; the *MEAN* of the IHS algorithm is 43.452, and the *STD* is 67.658. Both these values are greater than those of the PCA algorithm, indicating that the mean value of the IHS algorithm is moderate and its degree of distortion is better than that of the PCA algorithm. However, other indicators describing the image resolution, detail contrast, and texture transformation features are better than those of the IHS algorithm, but there are inconsistencies in the color information before and after fusion by the PCA algorithm. Overall, the PCA algorithm is better than the IHS algorithm. The *MEAN* of the Brovey algorithm is too low, only 13.690, which is not suitable, but its *RMSE* value is better than that of the PCA algorithm, and the remaining indicators have decreased to a certain extent. Similarly, the *MEAN*, *STD*, *AG*, and *EN* of the wavelet algorithm are better than those of the PCA algorithm. The larger their values, the better the fusion effect, and the color information before and after the fusion remains consistent; thus, the wavelet algorithm is better than the PCA algorithm. The *MEAN*, *STD*, and *EN* of the dual-transformation algorithm are 40.737, 65.525, and 0.979, respectively, and are slightly inferior to those of the wavelet algorithm; however, the image resolution, detail contrast, and texture transformation features are better than those of the wavelet algorithm. In summary, the dual-transformation algorithm has the best overall performance. The details of these results are shown in Figure 19.

Notably, the best quality of the multi-band fusion algorithm evaluation metrics is observed for color-NLRSIs from the dual-transformation algorithm. The relatively low values of *CC* and *EN* in the IHS, PCA, and Brovey algorithms indicate a serious spectral distortion problem in the images. The dual-transformation algorithm for image fusion outperforms other algorithms in terms of the information carrying capacity, detail characteristics, overall image clarity, and maintenance of color spectral information, indicating the best fusion quality. In summary, when comparing and analyzing three groups of images and five algorithms, the image fusion quality effect is in the following order: dual-transformation (IHS + Wavelet) > wavelet > IHS > PCA > Brovey. In summary, the IHS transform is simple and intuitive, but it may lead to spectral degradation and is not good for retaining spectral information. Wavelet transform can better capture the local features and detailed information of the image through wavelet decomposition and reconstruction. Meanwhile, the Brovey transform is a pixel-level fusion method, which is matched with the PAN light by weighting the wavebands. PCA transform is more efficient in fusion processing, but more sensitive to noise. In this study, we propose the dual-transformation method, which combines the advantages of the IHS and wavelet transforms; IHS transform provides processing of the spectral information, and wavelet transform extracts the detail and texture information of the image at different scales and also can better suppress the noise so that the result of fusion can be better retains the spectral information and spatial resolution. In addition, the methods of transforming IHS and wavelet can achieve adaptation to different image features by complementing each other’s advantages, which has enhanced the robustness of the dual-transformation algorithm to changes in image scales.

## 4. Conclusions and Discussion

In this study, a dual-transformation algorithm was proposed to generate high-quality color NLRSIs by fusing Landsat_MS, Landsat_PAN, and NPP-VIIRS-like datasets. The study area was Shanghai, China. Comparison and analysis with other single fusion algorithms was carried out to demonstrate the effectiveness of the dual-transformation algorithm. The dual-transformation achieved the best comprehensive performance effect in terms of spatial resolution, detail contrast, and color information after fusion, and the fused image was of the best quality. The accuracy of the color-NLRSI was improved greatly over the original NLRSI.

The experiment demonstrated the following: (1) The dual-transformation algorithm in this study had the best comprehensive performance in terms of spatial resolution, detail contrast, and color information before and after fusion. It also had the best fusion image quality, generating better color-NLRSIs; such images can be improved by 40 m in terms of their spatial resolution, compared with the original night light remote sensing images resolution of 500 m, and can distinguish the main distribution areas of different brightness. (2) Obtaining color-NLRSIs using the dual-transformation algorithm also solves the problems of low resolution and the small amount of feature information inherent in black and white night light images and visualizes the high radiation quality and true color of traditional black and white NLRSIs. These improvements have important research value for scientific research and data application of night light remote sensing images and can provide more accurate and detailed image information better to serve the needs of the development of human society. (3) Especially in the assessment of natural disasters, color-NLRSIs can not only obtain the regional human activity range but also observe the spatial information of features in the night light-covered area, which can be better used to analyze the regional spatial category information other than human activity information.

Therefore, the color-NLRSIs in this study can be used to comprehensively study small-scale regional carbon emissions and social-economic indicators, such as light pollution, through spatial-temporal analysis, which can help relevant departments formulate emission reduction policies and disaster prevention strategies according to local conditions.

## Figures and Tables

**Figure 1 sensors-24-00294-f001:**
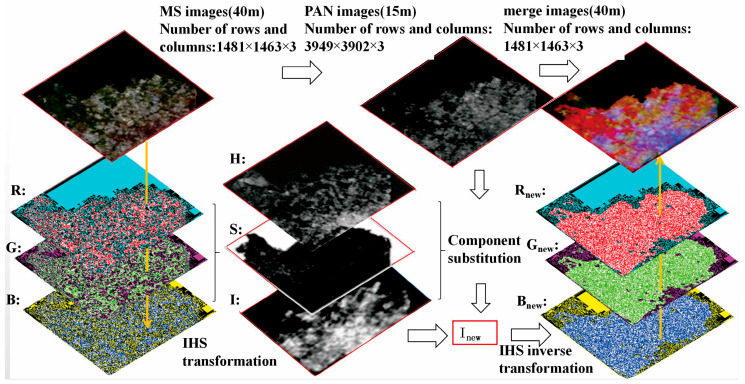
IHS color space adjustment from the dual-transformation.

**Figure 2 sensors-24-00294-f002:**
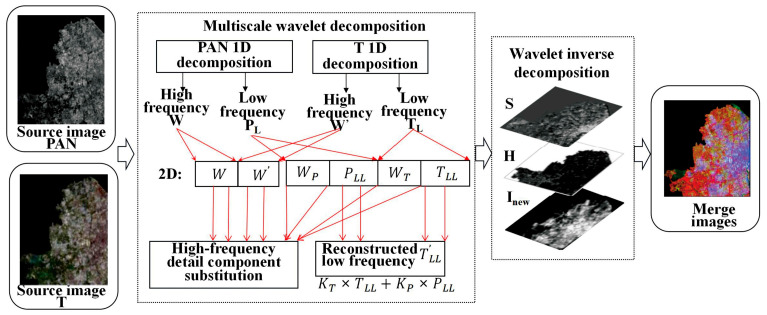
Wavelet transform of the dual-transformation.

**Figure 3 sensors-24-00294-f003:**
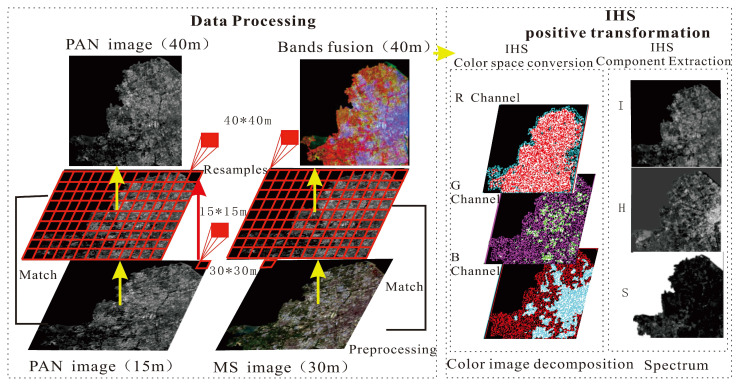
The dual-transformation of positive transform.

**Figure 4 sensors-24-00294-f004:**
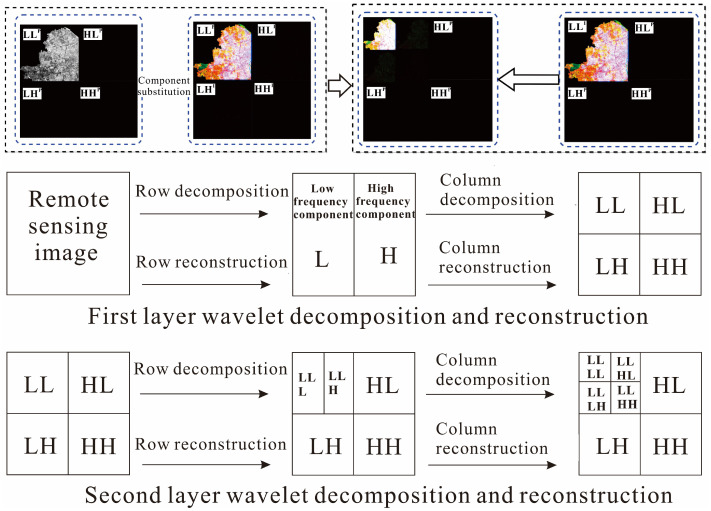
Two-dimensional wavelet decomposition improvement.

**Figure 5 sensors-24-00294-f005:**
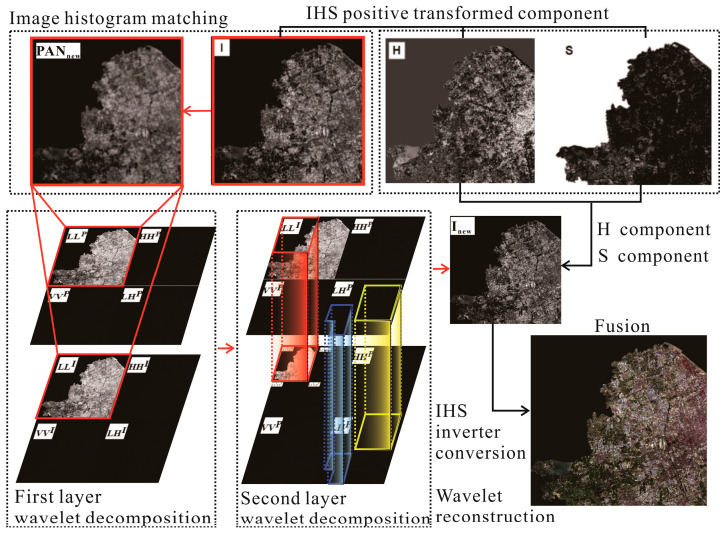
The dual-transformation of irreversible transform.

**Figure 6 sensors-24-00294-f006:**
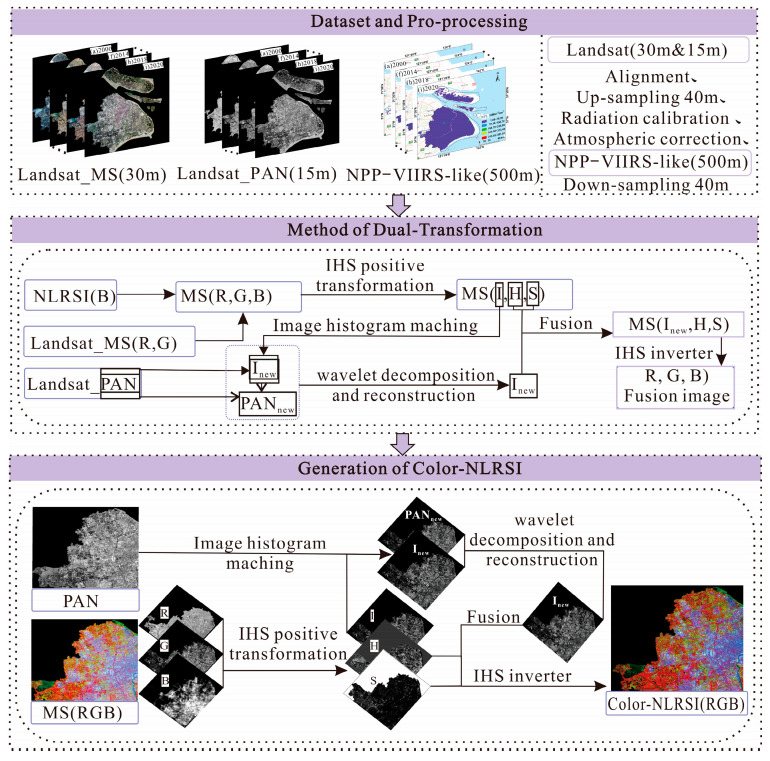
Generation of color-NLRSI based on dual-transformation.

**Figure 7 sensors-24-00294-f007:**
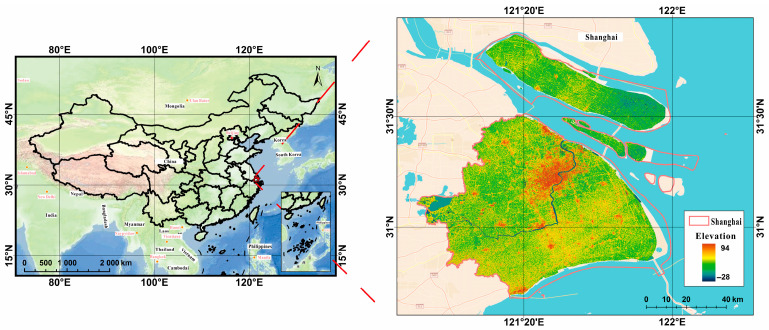
Study area of Shanghai in China.

**Figure 8 sensors-24-00294-f008:**
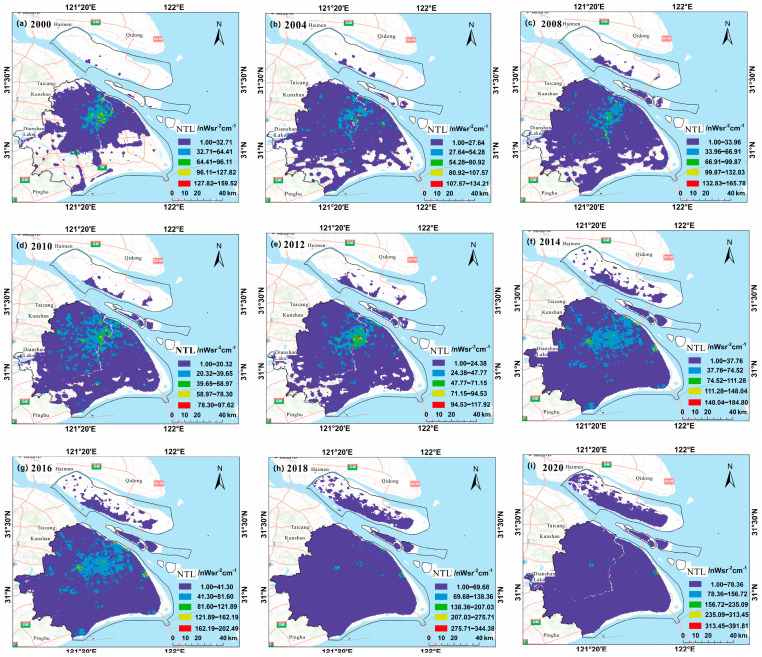
Pro-processing results of “NPP-VIIRS-like” NLRSI datasets in Shanghai.

**Figure 9 sensors-24-00294-f009:**
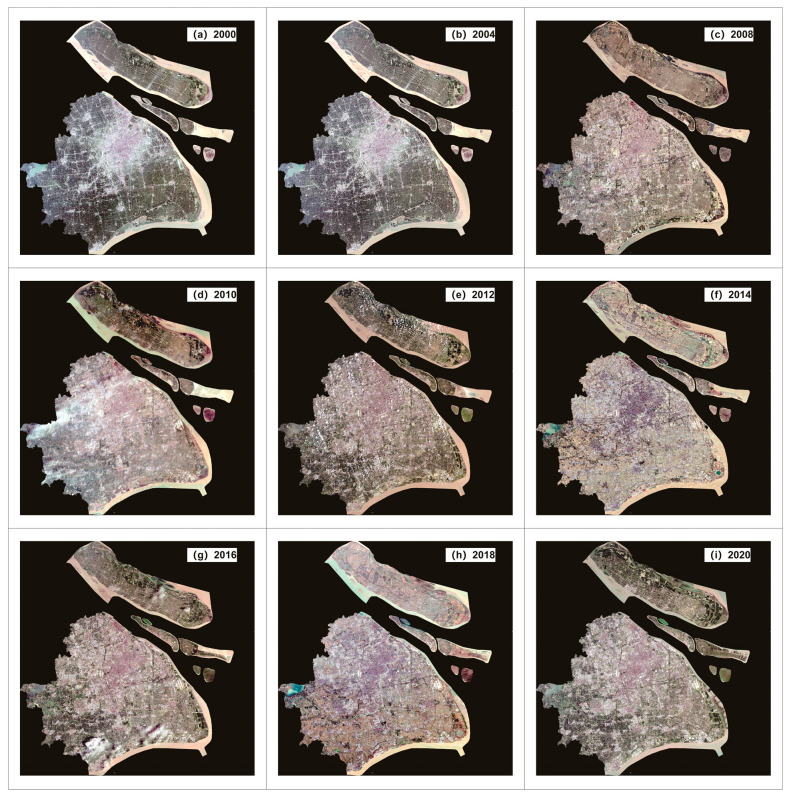
Pro-processing results of Landsat_MS datasets in Shanghai.

**Figure 10 sensors-24-00294-f010:**
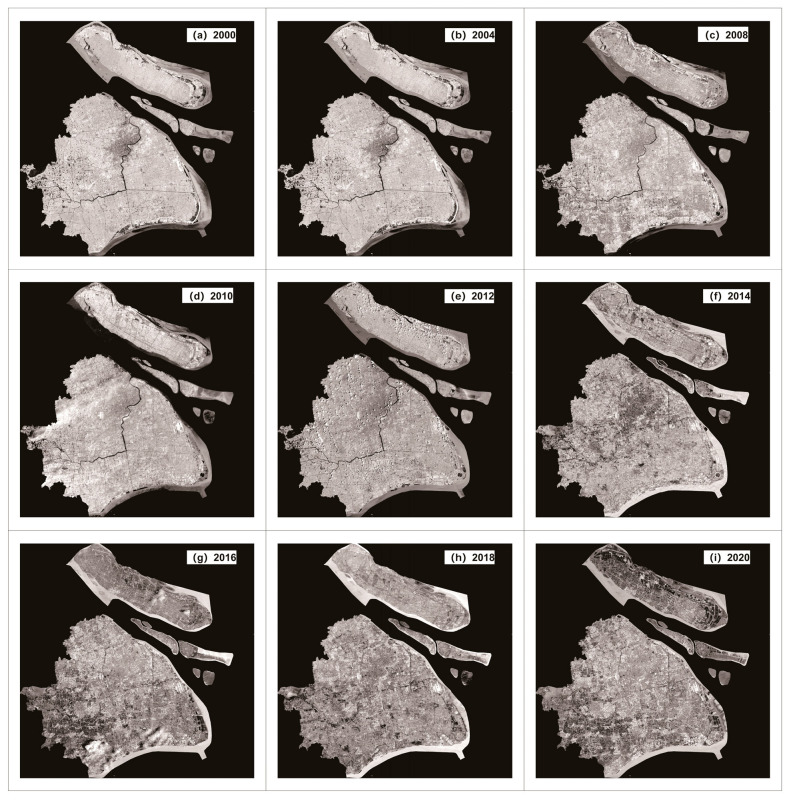
Pro-processing results of Landsat_PAN datasets in Shanghai.

**Figure 11 sensors-24-00294-f011:**
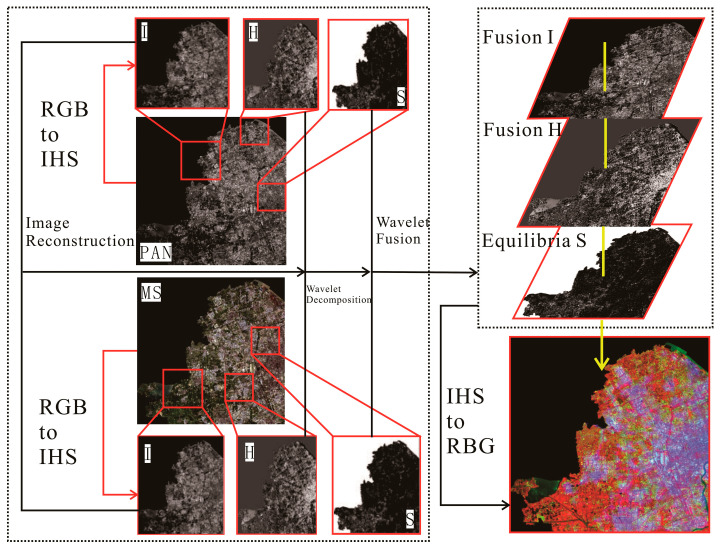
Fusion of the multi-source images using dual-transformation.

**Figure 12 sensors-24-00294-f012:**
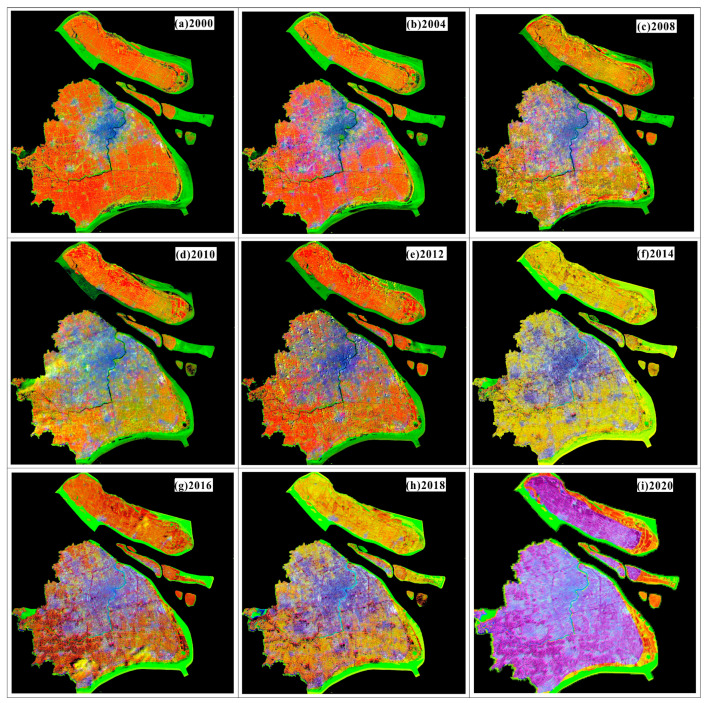
Generation of color-NLRSIs in Shanghai from 2000 to 2020.

**Figure 13 sensors-24-00294-f013:**
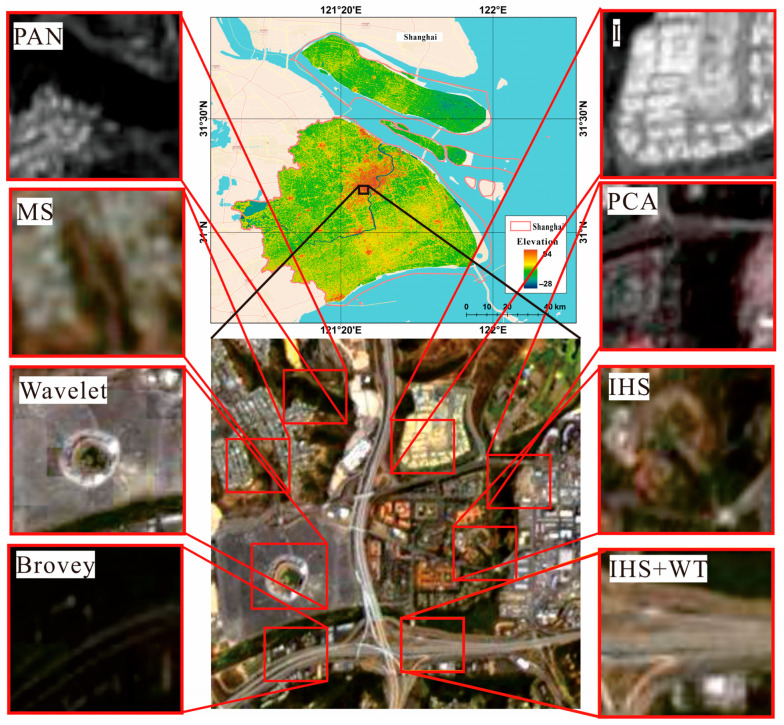
Comparison of different algorithms for Landsat images.

**Figure 14 sensors-24-00294-f014:**
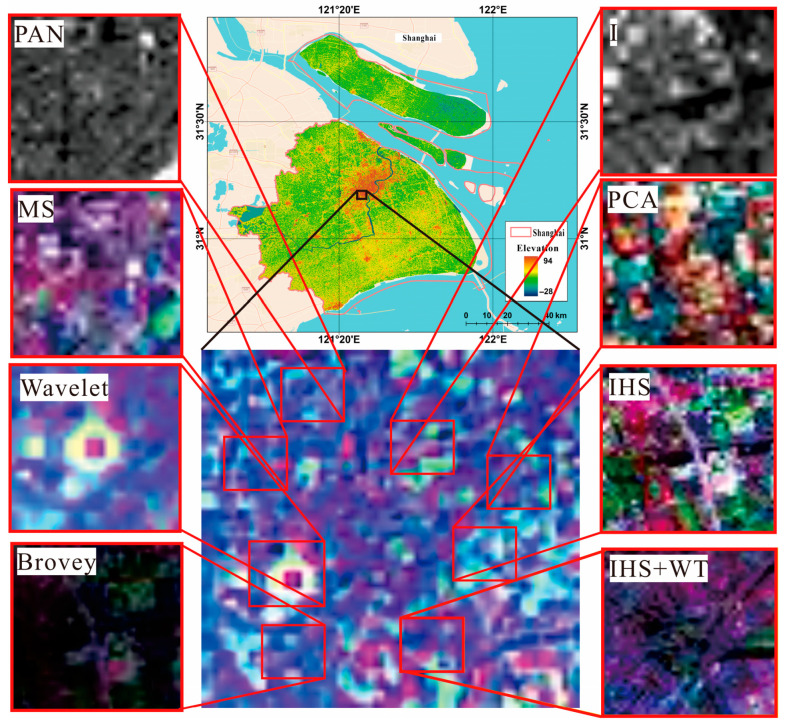
Comparison of different algorithms for color-NLRSIs.

**Figure 15 sensors-24-00294-f015:**
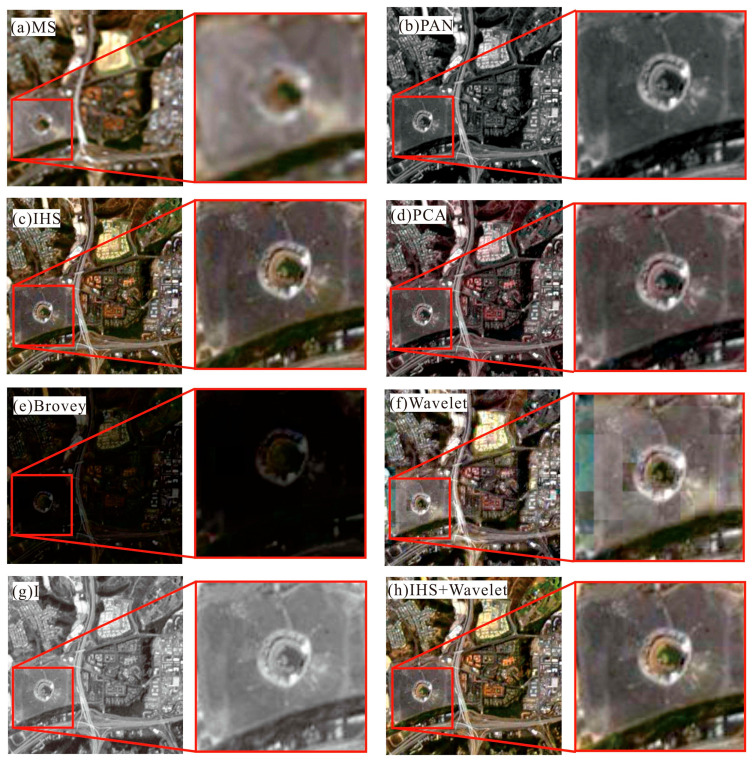
Comparison of different algorithms for fusion in Landsat_MS images.

**Figure 16 sensors-24-00294-f016:**
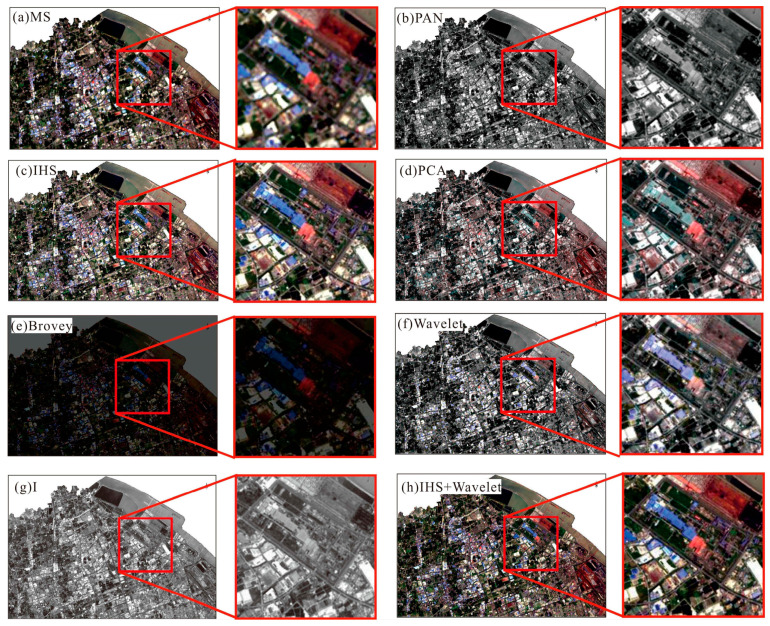
Comparison of different algorithms for fusion in Landsat_PAN images.

**Figure 17 sensors-24-00294-f017:**
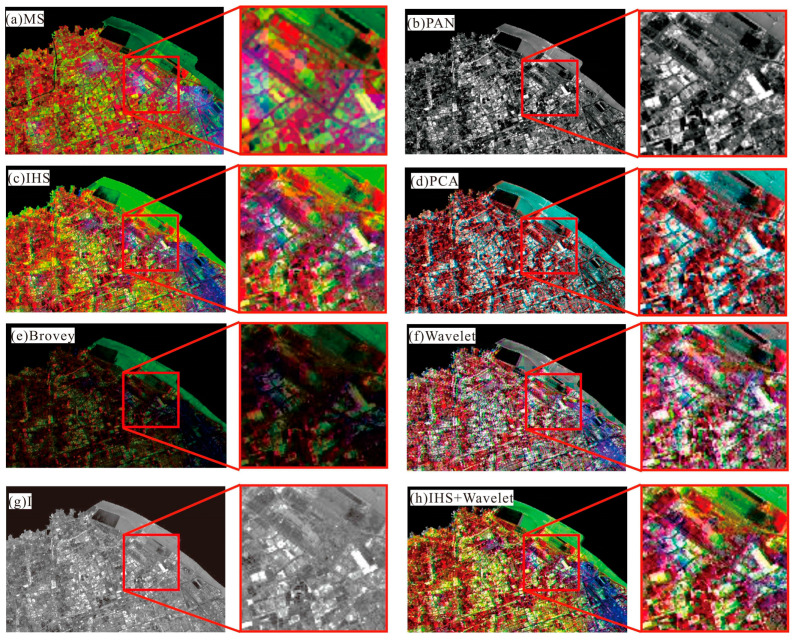
Comparison of different algorithms for fusion in color-NLRSIs.

**Figure 18 sensors-24-00294-f018:**
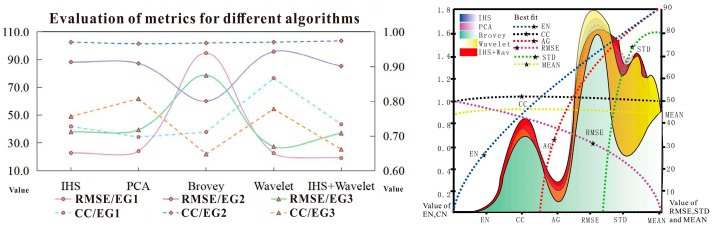
Quality evaluation of indicators.

**Figure 19 sensors-24-00294-f019:**
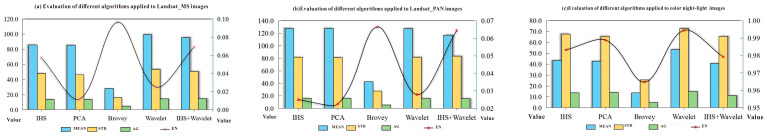
Evaluation of image quality metrics using different algorithms.

## Data Availability

No new data were created.

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
