# Peer review of "Color Night Light Remote Sensing Images Generation Using Dual-Transformation"

_sensors, 2024, doi:10.3390/s24010294_

Round 1

Reviewer 1 Report

Comments and Suggestions for Authors

In this manuscript, the authors proposed a fusion algorithm based on dual transform (wavelet transform and IHS (Intensity Hue Saturation) color space transform) to create color night light remote sensing images. The effectiveness of the method has been verified through experiments, but further adjustments are needed in the following aspects:

(1) The manuscript needs to provide the entire process of the method;

(2) The method in the manuscript should be very sensitive to alignment, and additional explanation is needed in this section;

(3) The method in the manuscript needs to further explain the selected data, as the time span of one year is obviously too large;

(4) The figures in the manuscript are very unclear and not easy to read;

(5) The comparative experiments in the manuscript need to highlight two points. Firstly, different methods should try to select the same area for comparison, and secondly, more night light remote sensing images should be selected for comparison.

(6)There are some obvious English errors in the manuscript that require careful revision.

Comments on the Quality of English Language

Require careful revision

Author Response

Dear reviewer:
    Our revisions are attached.

Reviewer 2 Report

Comments and Suggestions for Authors

-> The paper introduces equations (1)-(3) for IHS transformation, but the notation and explanation lack clarity. Provide a more detailed explanation of the variables and their significance in these equations.

-> Figure 1 is referenced in the text, but there's no detailed explanation of the components involved in the IHS color space transformation. Enhance the figure caption or provide additional details in the text to aid comprehension.

-> The paper mentions that IHS transformation retains spatial structure information to a greater extent, but lacks a clear justification for choosing this method. 

-> The paper mentions that IHS transformation may lead to spectral degradation. Elaborate on this point, quantify the degree of degradation, and discuss how it impacts the results.

-> The paper suggests integrating other algorithms with IHS transformation for better multi-band image fusion. Provide more details on the rationale behind this integration and how it addresses the limitations of IHS transformation alone.

-> The section on wavelet transform lacks a step-by-step explanation. Provide a clearer description of the process, including the significance of equations (5) and (6).

-> The paper introduces fusion coefficients without a clear justification for the chosen methods. Explain the rationale behind selecting the maximum absolute coefficient method, weighted average method, and local variance criterion.

-> The paper lacks a comparative analysis with existing image fusion methods. Include a discussion on how the proposed approach compares to other state-of-the-art techniques in terms of accuracy, efficiency, and robustness.

->The paper introduces various parameters, such as weighting factors in equation (7) and fusion coefficients. Provide an analysis of how changes in these parameters affect the results to ensure robustness.

->The paper briefly mentions experiments in Shanghai but lacks details on the experimental setup, data sources, and validation metrics. Provide a comprehensive description of the experiments, including validation methods and results.

Comments on the Quality of English Language

There are many long and confusing lines.

Author Response

(The authors gave the same response as above.)
